# Free Radicals and ROS Induce Protein Denaturation by UV Photostability Assay

**DOI:** 10.3390/ijms22126512

**Published:** 2021-06-17

**Authors:** Paolo Ruzza, Claudia Honisch, Rohanah Hussain, Giuliano Siligardi

**Affiliations:** 1Padova Unit, Institute of Biomolecular Chemistry of CNR (ICB-CNR), Via F. Marzolo, 1, 35030 Padova, Italy; claudiahonisch@gmail.com; 2Department of Chemical Sciences, University of Padua, Via F. Marzolo, 1, 35030 Padova, Italy; 3Diamond Light Source Ltd., Harwell Science and Innovation Campus, Didcot OX11 0DE, UK; rohanah.hussain@diamond.ac.uk

**Keywords:** VUV and far-UV irradiation, protein photo-stability, synchrotron radiation circular dichroism, reactive oxygen species, UV protein denaturation assay

## Abstract

Oxidative stress, photo-oxidation, and photosensitizers are activated by UV irradiation and are affecting the photo-stability of proteins. Understanding the mechanisms that govern protein photo-stability is essential for its control enabling enhancement or reduction. Currently, two major mechanisms for protein denaturation induced by UV irradiation are available: one generated by the local heating of water molecules bound to the proteins and the other by the formation of reactive free radicals. To discriminate which is the likely or dominant mechanism we have studied the effects of thermal and UV denaturation of aqueous protein solutions with and without DHR-123 as fluorogenic probe using circular dichroism (CD), synchrotron radiation circular dichroism (SRCD), and fluorescence spectroscopies. The results indicated that the mechanism of protein denaturation induced by VUV and far-UV irradiation were mediated by the formation of reactive free radicals (FR) and reactive oxygen species (ROS). The development at Diamond B23 beamline for SRCD of a novel protein UV photo-stability assay based on consecutive repeated CD measurements in the far-UV (180–250 nm) region has been successfully used to assess and characterize the photo-stability of protein formulations and ligand binding interactions, in particular for ligand molecules devoid of significant UV absorption.

## 1. Introduction

Biotherapeutics are becoming the mainstream of new medicinal agents, from which monoclonal antibodies and peptides are of great pharmaceutical interest. The development of biopharmaceuticals is often hampered by the reduced or lack of stability during ageing under a variety of environmental factors such as temperature, light, and oxidation that is manifested by a loss of ordered structure or protein misfolding mirrored by the loss or change of function. Photo-stability can be an issue in the development and formulation of biopharmaceuticals [1].

Circular dichroism (CD) spectroscopy is the ideal technique to characterize and monitor the folding of protein in solution as a function of environmental factors such as temperature, pH, solvent polarity, salts, detergents, lipids, and ligand interactions [2,3,4]. Unlike macromolecular crystallography (MX) and NMR where detailed structural models of the proteins at atomic resolution can be achieved, CD spectroscopy provides a fast method to analyze the conformational behavior in solution to confirm that different environmental conditions more appropriate for MX and NMR measurements do not affect the protein folding observed under physiological conditions. There are many techniques that can be used to determine binding interactions such as fluorescence, isothermal titration calorimetry (ITC), and surface plasmon resonance (SPR), however, CD spectroscopy is the only technique that can reveal directly what type of protein conformational change might be induced upon ligand addition. Is the protein folding retained or does it increase the content of one element of secondary structure such as α-helix, β-strand, β-turn, or unordered at the expense of others? Certainly, this question cannot be addressed directly with fluorescence, ITC, or SPR techniques.

Although bespoke bench-top CD spectropolarimeters using Xe lamp as the light source can operate in the far-UV-Visible spectral region (175–700 nm), the use of synchrotron light sources can be extended to the lower wavelengths in the Vacuum UV (VUV) region (130–200 nm) for the measurements in the solid state of thin films [5]. In the vacuum and far UV spectral regions (125–250 nm), the photon-flux and brilliance of synchrotron light sources compared to that of Xenon lamps is increased substantially [6,7], which means the signal-to-noise ratio of synchrotron radiation circular dichroism (SRCD) spectra have also increased.

The high photon flux and brilliance in the vacuum UV (VUV) and far-UV region of Diamond B23 beamline can induce protein denaturation on scanning repeated consecutive SRCD spectra. This can be eliminated, however, by reducing the slit width from 0.500 mm to 0.200 mm corresponding to a bandwidth of 1.2 nm and 0.5 nm, respectively [8], or to rotate the cuvette cell around the incident light propagation using a motorized rotating cylindrical cell holder (Figure 1).

However, when UV denaturation is produced, a significant decrease of secondary structure is observed for proteins with α-helical and/or β-sheet conformations. Upon scanning repeated consecutive SRCD spectra in the 185–250 nm region, the rate of the conformational changes induced by the high photon flux of the light source has been found to be dependent on the protein primary sequence as well as the solution environment such as solvent composition, pH, temperature, protein concentration, and chemical agent [7,9,10,11]. This has led to the development at Diamond B23 beamline of a protein UV denaturation assay by simply conducting consecutive repeated CD measurements in the 180–250 nm region to be used as a method to assess the relative photo stability of proteins as a function of their formulations (Figure 2). As the rate of UV denaturation is very sensitive to the radiation power of the irradiating light source and the dose of irradiation, it is recommended to determine the number of repeated consecutive scans with the equipment using human serum albumin essentially fatty acid and immunoglobulin free (HSAff) as the control (Figure 2C). Depending on the radiation power, for example with B23, 20 scans are sufficient, whereas with Chirascan CD instrument with 4 nm bandwidth at least 50 repeated scans are required to induce a significant HSAff denaturation that can be used as a parameter to investigate the proteins of interest. This can be used to grade the protein photostability with respect to albumin under similar environmental conditions.

Interestingly, the relative rate of UV denaturation was found to be significantly affected by ligand binding interactions [7,8,9,10,11] (Figure 2A–C). This qualitative assay turned out to be very useful to determine binding interactions of molecules with negligible UV absorption like for drugs devoid of aromatic or π-conjugated moieties, saturated lipids, and sugars that are otherwise challenging systems to be studied by other methods and techniques (Figure 2A–C).

Similar to thermal denaturation, UV denaturation varies from protein to protein, showing different conformational changes that may be correlated with the degree of protein stability (Figure 2E). Different mechanisms to explain this phenomenon have been proposed, involving the generation of oxygen free radicals and other non-radical reactive oxygen species from the aqueous solution medium [12], the heating of internal bound water molecules implicated in maintaining the protein native structure [13], or both.

The aim of this study is to determine whether the protein UV denaturation is due to the local heating from irradiated protein bound water molecules or from the formation of free radical products. The understanding of the origin of the protein denaturation by irradiating the sample in the far-UV region with high photon flux light sources can also be used as a UV protein denaturation assay to assess and characterize the photo-stability of protein formulations and ligand binding interactions, in particular for ligand molecules devoid of or with negligible UV absorption in the far-UV region.

## 2. Results and Discussion

### 2.1. Assessment of the Hypothesis That Protein Denaturation Induced by UV Irradiation Is Originated by the Heating of Protein Bound Water Molecules

The heating hypothesis was assessed by comparing the effects of heating from 5 °C to 90 °C against the far-UV irradiation induced at 23 °C by scanning 50 repeated consecutive SRCD spectra of two aqueous solutions of human serum albumin fatty acid and essentially globulin free (HSAff) at 5 µM and 10 µM concentration, respectively, repeated for each denaturation type.

In Figure 3, the results of thermal (Figure 3A–F) and UV-denaturation (Figure 3G–I) of 5 µM and 10 µM HSA aqueous solutions by CD spectroscopy are compared head-to-head. The melting temperature of both HSA solutions remained unchanged (Figure 3C,D) with the identical first derivative at 60 °C (Figure 3E,F) unlike the rates of UV-denaturation (Figure 3I), which clearly indicated significant differences that were quantified by the rates of changes using the exponential fitting equation.

For each CD spectrum, the secondary structure estimation (SSE) calculated using the CONTINLL algorithm [17] showed an increased content of β-strand and β-turns at the expense of the α-helix content (Figure 4). Both thermal denaturation assays were partially reversible with 86% recovery of the α-helix content when cooled back to room temperature (Table 1).

For the 5 µM and 10 µM HSAff aqueous solutions, the UV irradiation experiment conducted by scanning 50 repeated consecutive SRCD spectra at constant 20 °C using 2 nm bandwidth was found to be concentration dependent (Figure 3G,H) revealing different rates of spectral changes (Figure 3I). Similar to the thermal denaturation analysis, the SSE calculated for each SRCD spectrum was displayed for each element of the secondary structure (Figure 4). The fitting of the CD data at 191 nm using the exponential decay equation described in the Materials and Methods section showed a rate of protein denaturation to be about 20 times bigger for the diluted 5 µM HSAff (k = 0.0855 s^−1^) solution than that of 10 µM (k = 0.0041 s^−1^). This implies that the relative difference in protein denaturation rates induced by UV irradiation being concentration sensitive is diffusion dependent. For both denatured states of the 50th spectrum, there is an increased viscosity compared to the corresponding first spectrum (Figure 3I) that is consistent with the observations reported in Davies et al. [18] for protein aqueous solutions when irradiated with UV light. Another important difference between the two types of experiments was that the thermal denaturation was partially reversible (Figure 5 and Table 1) whereas the UV photo-denaturation was completely non-reversible.

If the local heating hypothesis of bound water to the protein proposed by Wien et al. (2005) [13] were correct, the effects of UV irradiation on both HSAff solutions of 5 µM and 10 µM should be partially reversible. As this was not the case, the heating hypothesis was not the mechanism for protein denaturation when UV irradiated.

Although both protein thermal and UV denaturation showed an increase in β-strand content mainly at the expense of the decreased α-helical content, this occurred in different pathways (Figure 4 and Table 1).

### 2.2. Assessment of the Hypothesis That the Protein Denaturation Induced by UV Irradiation Is Originated by the Formation of Free Radicals

Electron spin resonance and paramagnetic resonance (ESR or EPR) are techniques that can detect specifically and directly the presence of free radicals (FRs). The use of spin trapping probe can compensate for the relatively low sensitivity using aqueous solutions at room temperature. However, due to the requirement of specialized and expensive ESR spectrometer, alternative methods have been developed for the FRs detection with more readily available equipment based on the detection of FRs reaction products using a variety of probe molecules (for exhaustive reviews, see Refs. [19,20]). The best and simplest probes are molecules with optical properties that change after reacting with FRs. In particular, fluorescent probes permit the detection of FRs with higher sensitivity compared to other spectroscopic probes. From the fluorescent probes available, especially useful are the ‘‘positive’’ fluorogenic probes. These probes are non-fluorescent or very weakly fluorescent molecules that become substantially fluorescent upon reaction with FRs. In this study, we used dihydrorhodamine 123 (DHR-123), a non-fluorescent molecule in its initial state, that is converted to the highly fluorescent form Rhodamine 123 (Rh-123) upon reaction with free radicals (Appendix A) [21].

The capability of the high photon flux of B23 beamline in the far-UV region to generate free radicals in aqueous solutions was assessed by fluorescence spectroscopy using a fluorescent sensor. Initially, the generation of FRs was evaluated by fluorescence spectroscopy after irradiation at 254 nm with the UV-C lamp BioLink 254 photo-reactor whilst the effect of UV exposure to the protein conformation was monitored using the benchtop Chirascan Plus CD instrument with 1 nm bandwidth that did not denature the investigated protein as a result of UV irradiation on scanning the CD spectra (Figure 6). For this assay, an aqueous solution of DHR-123 was kept under UV-C exposure at 254 nm at room temperature for different amounts of time, from 0 to 60 s. The corresponding fluorescence emission spectra in the 510–700 nm range when excited at 505 nm were measured for the UV-C irradiated and non-irradiated solutions as control. In Appendix A, the fluorescence of DHR-123 in aqueous PBS buffer solution was found to increase as a function of UV-C irradiation. The rate of fluorescence change (Appendix A, insert, black line) represented the rate of conversion of the non-fluorescent DHR-123 into the fluorescent Rh-123 (Appendix A), which reflected the rate of free radical formation.

Since the non-fluorescent DHR-123 can be oxidized via photosensitization, a parallel experiment was conducted in the presence of 0.1 mM ascorbate, a known free radical scavenger [22]. Under this condition, the fluorescence emission was detected after UV-C irradiation (Appendix A, insert, blue line). These results indicated that the transformation of DHR-123 in the fluorescent Rh-123 was due to the action of free radicals generated by the UV irradiation of the aqueous buffer solution and not by a photo-oxidation process. At micromolar solute concentrations, free radicals are predominantly produced from water molecules due to their higher concentration (55 M) [20] involving many different reactive oxygen species (ROS), including HO^●^, HO_2_^●^, O_2_^●−^. These free radicals have limited lifetimes and limited diffusion ranges, for example, a few nanometers for the most abundant hydroxyl free radical [23,24], hence they are considered to be localized on the site of irradiation.

As a control, the fluorescence emission of DHR-123 was also measured as a function of temperature. Ramping between 5 and 85 °C did not produce any significant increase in the fluorescence emission, as shown in Appendix A. This result suggests that the increase in the DHR-123 maximum fluorescence emission at 524 nm observed when the sample is irradiated with the consecutive repeated SRCD spectra is related to the generation of ROS in the aqueous media and is attributable only to the exposure to UV light and not to the heating of water molecules, whether bound to the protein or free.

DHR-123 in the presence of ovalbumin (OVA) showed an enhanced fluorescence at 524 nm (Figure 6) being consistent with an increased production of free radicals. The initial rate of change in the absence or presence of the protein rose from 0.245 to 0.302 sec^−1^ respectively. This may be due to the injection of electron from the side chain of UV-excited aromatic residues (Trp, Tyr and Phe), which can be captured by both O_2_ present in the aqueous solution and by disulphide bridges, leading to the formation of HO_2_^●^ or O_2_^●^ and disulphide electron adduct radicals, respectively [19]. As the addition of protein increases, the viscosity of the solution decreases the motion of molecules favoring the interaction of photons with water molecules as well as ROS with the non-fluorescent probe DHR-123.

The conformational effect on the secondary structure of ovalbumin due to UV irradiation was evaluated by CD spectroscopy under the same parameters of protein concentration and cuvette pathlength of the fluorescence experiments. The far-UV CD spectra of ovalbumin as a function of UV irradiation up to 900 s were qualitatively diagnostic of protein denaturation with loss of α-helical content.

These results indicated that the mechanism of UV-photo denaturation of a protein was due to the formation of ROS from the aqueous medium that denatured the protein.

The reaction of protein with ROS may occur via hydrogen abstraction from saturated carbon or hydroxyl addition to unsaturated double bonds or aromatic rings [25]. The H-abstraction is dependent upon the single bond strength (the bond energies for C-H, N-H, O-H, and S-H are 411, 386, 459, and 363 KJ/mol, respectively, at 25 °C) and influenced by the electron properties of the substituent: an electron-donating substituent increases the reactivity, while an electron-withdrawing decreases it. The S-H bond, having the lowest bond energy, makes cysteine residue one of the most reactive moiety for the H-extraction. At neutral pH, the amine group that is positively charged and thus electron-deficient would not be subjected to a direct attack of free radicals. Neighboring atoms or groups able to stabilize the nascent radicals will make the radical attacks more likely. Hydroxyl radicals attack preferentially the side chain of solvent-exposed amino acid residues due to the higher accessibility compared to the buried or less exposed backbone chain. This may oxidize the amino acid side chains of key residues and promote a loss of ordered structure without the formations of protein fragments. An extensive description of these two processes is available on Neves-Petersen et al. [19,20]. This was consistent with the observation that the gel page of irradiated and not irradiated proteins showed no detectable protein backbone cleavage upon far-UV irradiations [13], similar to the experiments discussed here.

The fact that only four consecutive repeated SRCD spectra in the far-UV region (185–260 nm) at constant 23 °C were sufficient to induce a change in the fluorescence spectrum of DHR-123 (Appendix A) indeed demonstrated the production of ROS when the sample was irradiated with the powerful B23 light source.

## 3. Materials and Methods

Dihydrorhodamine 123 (DHR-123), ovalbumin (S7951), human serum albumin fatty acid, and globulin free (HSAff) (A3782), and DMSO were purchased from Sigma-Aldrich (Milan, Italy) and used without any other treatment.

Fluorescence experiments were carried out using a Chirascan-Plus CD Spectrometer with fluorescence attachment (Applied Photophysics Ltd., Leatherhead, UK). Briefly, 1.5 μL of DHR-123 in DMSO (2 mg/mL) were added to 3000 μL of 20 mM PBS, pH 7.4, in a Suprasil fluorescence cell with 1.0 cm path length and irradiated using BioLink 254 photoreactor (Vilber, Eberhardzell, Germany). The fluorescence emission spectrum of irradiated DHR-123 solution in the 510–700 nm range (Ex 505 nm, slit 4 nm) was recorded at different irradiation time. The emission spectra of the not irradiated solution was recorded in a temperature range between 5 °C and 85 °C at 10 °C increments wit 8 min incubation time for equilibration.

The protein denaturation by heating was monitored by CD spectroscopy using Chirascan Plus (Applied Photophysics Ltd., Leatherhead, UK) with 1 nm bandwidth that did not promote any protein UV denaturation in each of the repeated scans at an increased temperature from 5 °C to 90 °C at 5 °C increments with 5 min incubation time.

The UV irradiation was conducted at Diamond Light Source synchrotron (Harwell Science and Innovation Campus, Didcot, UK) using beamline B23 module B for SRCD at constant 23 °C and analyzed with CDApps suite of programs [26]. It is important for the investigated protein systems to keep the same number of repeated scans for the same wavelength range at the same scan speed. For example, 50 repeated consecutive spectra in the 178–255 nm wavelength range took 150 min to be completed, which was similar to that for the thermal denaturation.

The rates of denaturation from CD and SRCD data at 191 nm were calculated fitting the exponential decay equation [y = y_0 +_ Ae^(−x/t)^] (ExpDec1 fit of Origin (OriginLab)) where A = amplitude, t = time constant and y0 = offset. The equation used for the rate of denaturation was k = 1/t (s^−1^).

## 4. Conclusions

Peptides and proteins as biotherapeutics are mainstream new medicinal agents, whose development is often hampered by the lack or reduced stability during ageing under a variety of environmental factors such as temperature, light, and oxidation.

The protein UV denaturation assay developed at Diamond B23 beamline for SRCD resulted in a facile, accurate, and fast assessment of the relative protein photo-stability as a function of environment, such as solvent/buffer composition, pH, red-ox, and surfactants to screen the conditions to enhance protein photo-stability. It can be used to qualitatively assess the protein binding interactions of UV transparent ligands or with negligible absorption such as lipids, sugars, and metal ions.

The aim of this study was to demonstrate that the UV protein denaturation was not due to thermal effects but to free radicals. To unambiguously determine the origin of protein denaturation, UV-irradiation experiments to induce the denaturation of albumin and ovalbumin proteins by heating and UV irradiation at constant room temperature were performed.

We have demonstrated the hypothesis that protein denaturation induced by UV irradiation is originated by the local heating of the molecules of water bound to the protein, as proposed by Wien et al. [13], is not correct. At first glance, the comparison of the two methods of denaturation: one by heating from 5 to 90 °C and the other by scanning with B23 beamline 50 repeated consecutive spectra in the 178–255 nm region and both performed within the same length of time, about 150 min, revealed apparent spectral similarity. However, several major differences were observed between the two methods. The thermal denaturation was found to have degrees of reversibility and was protein concentration independent, whereas the UV denaturation was irreversible and protein concentration dependent with substantial differences in the rates of denaturation. Finally, the protein unfolding induced by UV irradiation and heating was occurring under distinct paths with different amount of the secondary structure estimated from CD data. Indeed, for a given protein, in this case the HSA, the detailed analysis of the thermal and UV denaturation experiments showed unambiguously substantial differences, as illustrated in Figure 3.

We determined that the origin of protein denaturation by UV irradiation was solely due to the free radicals-ROS formation revealed by fluorescence spectroscopy using the temperature insensitive photosensitizer DHR-123, as shown in Figure 6. The fact that DHR-123 was becoming fluorescent when converted into Rh-123 only when irradiated at single wavelength (254 nm) using UV lamps or scanning four consecutive repeated SRCD spectra in the far-UV region using B23 beamline, and not by heating (Appendix A), was unambiguously indicative that the denaturation of aqueous proteins was indeed promoted by the reactive FRs-ROS species and not by the heating effect on the water molecules bound to the protein when UV irradiated.

The scan speed of the measurement dictates the irradiation time for the consecutive repeated scans. In this manner the irradiation time can be quantified. Of course, the SRCD measurement induces a protein conformational change while monitoring it due to the high photon flux. However, as the irradiation time, the spectral range, and scan speed are known, the rate of denaturation is quantifiable.

This manuscript is not a comparison of the UV photo-stability of proteins but rather an assay to assess the relative photo-stability of protein under a variety of environmental conditions such as solvent polarity, ionic strength, and ligand binding interactions.

In summary, protein denaturation induced by FRs and ROS promoted by irradiation at 254 nm using UV lamps or by repeated consecutive SRCD spectra is a facile, accurate and fast method to assess the protein conformation stability and qualitatively the binding interaction of transparent ligands that is distinct and complementary to the thermal denaturation method.

## Figures and Tables

**Figure 1 ijms-22-06512-f001:**
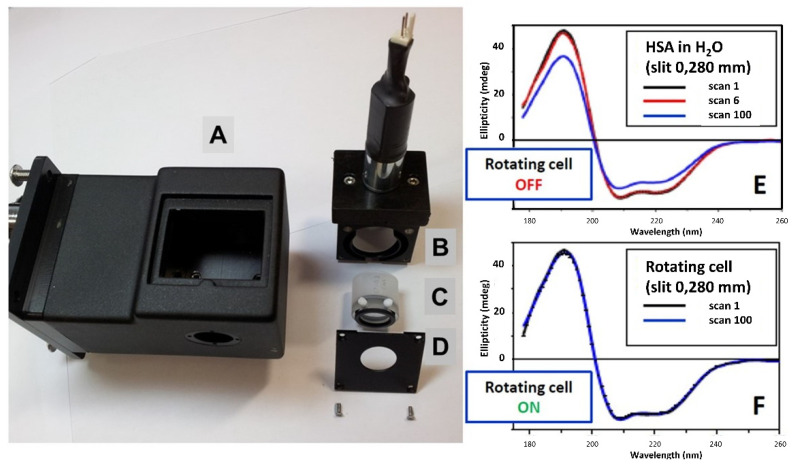
Motorized rotating cuvette cell holder compatible with Peltier temperature stage (5 °C to 90 °C) to eliminate in this example the protein denaturation effect from UV irradiation by up to 100 consecutive repeated scans using Diamond B23 beamline for SRCD spectroscopy. (**A**) Peltier temperature stage (Quantum Northwest, Liberty Lake, WA, USA). (**B**) Motorized rotating cylindrical cell holder. (**C**) Cylindrical cuvette cell (Hellma, Southend on Sea, U.K.). (**D**) Flat cover to enclose and enable the cuvette C to be rotated inside the cell holder B. (**E**) One hundred consecutive repeated CD spectra of aqueous human serum albumin (HSA) using a 0.280 mm slit width corresponding to 0.7 nm bandwidth measured with the rotating cell holder switched OFF. For clarity, only the 1st, 6th, and 100th scans are reported to illustrate the spectral changes induced by the UV irradiation. (**F**) One hundred consecutive repeated CD spectra measured under continuous rotation with the motorized rotating cell holder switched ON. The 100 spectra of HSA are superimposed, indicating no protein denaturation after 100 consecutive repeated scans.

**Figure 2 ijms-22-06512-f002:**
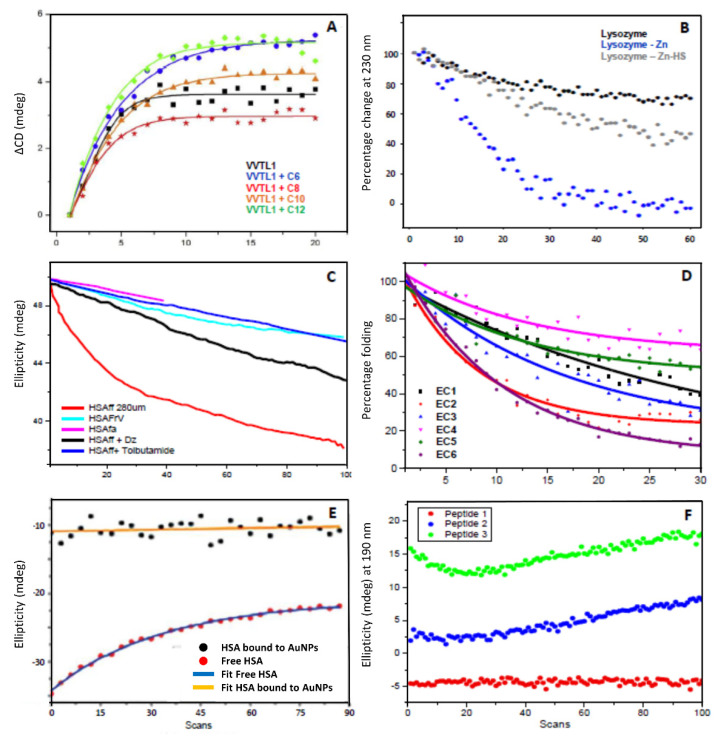
Examples of UV denaturation studies on different proteins. (**A**) Plot of SRCD intensity change (ΔCD) at 195 nm for Thaumatin-like protein VVTL1 in the presence of ethyl hexanoate (C6, blue), ethyl octanoate (C8, orange), ethyl decanoate (C10, red), and ethyl dodecanoate (C12, green) and absence of ethyl esters (black) calculated by subtracting the CD value of scan 1 from all the consecutive scans. Redrawn from [9]. (**B**) Plot of repeated consecutive SRCD signal at 230 nm for lysozyme with and without ligands showed conformational changes with different rates of protein denaturation. Redrawn from [14]. (**C**) Rate of UV protein unfolding (denaturation) of human serum albumin fatty acid free (HSAff) in H_2_O with and without ligands such as fatty acid (octanoic acid), diazepam, and tolbutamide plotted at 190 nm for 100 repeated consecutive SRCD spectra. Redrawn from [15]. (**D**) Rate of UV protein denaturation of antibody Mab-1 in different formulations assessed with 30 repeated consecutive SRCD spectra. Redrawn from [11]. The rates reported in percentage of protein folding were calculated by dividing the CD intensity at a fixed wavelength of the protein-ligand complex by that of the protein alone at the same concentration as that of the complex and multiplied by 100. (**E**) Plot of SRCD signal at 209 nm as a function of UV-exposure time that is equivalent to light irradiation when scanning repeated consecutive SRCD spectra for free human serum albumin HSA (red circles for experimental data and blue line for fitting) and for HSA bound to AuNP (black squares for experimental data and orange line for fitting). Redrawn from [16]. (**F**) Plot of repeated consecutive SRCD signal at 190 nm of three related vasoactive intestinal peptide (VIP) in MES buffer: a wild type (peptide 1, red), one mutated W25S (peptide 2, blue) and one mutated W25S with palmitoylated K20 (peptide 3, green). Redrawn from [11].

**Figure 3 ijms-22-06512-f003:**
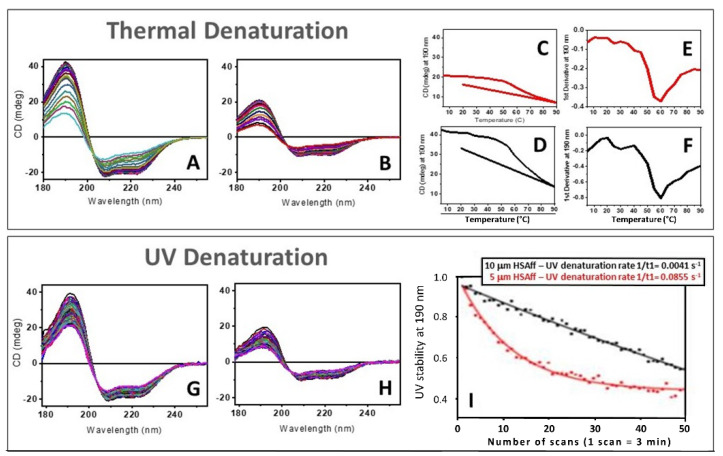
Comparison between thermal and UV denaturation of two solutions of human serum albumin essentially fatty acids and globulin free (HSAff) at different concentrations. (**A**,**B**) Thermal denaturation experiments: CD spectra of aqueous solutions of 10 µM and 5 µM HSAff as a function of temperature from 5 °C to 90 °C at 5 °C interval and back to 20 °C to assess reversibility. (**C**,**D**) Plots of CD intensity at 190 nm versus temperature and the corresponding first derivative plots (**E**,**F**). (**G**,**H**) UV denaturation experiments: SRCD spectra of aqueous solutions of 10 µM and 5 µM HSAff as a function of 50 consecutive repeated scans. (**I**) Plot of the normalised SRCD intensity at 190 nm calculated by dividing the value of each scan by that of the first one versus the number scans. The plots represent the rate of protein denaturation by UV irradiation. The experimental data (black squares for HSAff 10 µM and red squares for HSAff 5 µM) were fitted with an exponential decay equation [y = y_0 +_ Ae^(-x/t)^] (ExpDec1 fit using Origin suite of programs (OriginLab).

**Figure 4 ijms-22-06512-f004:**
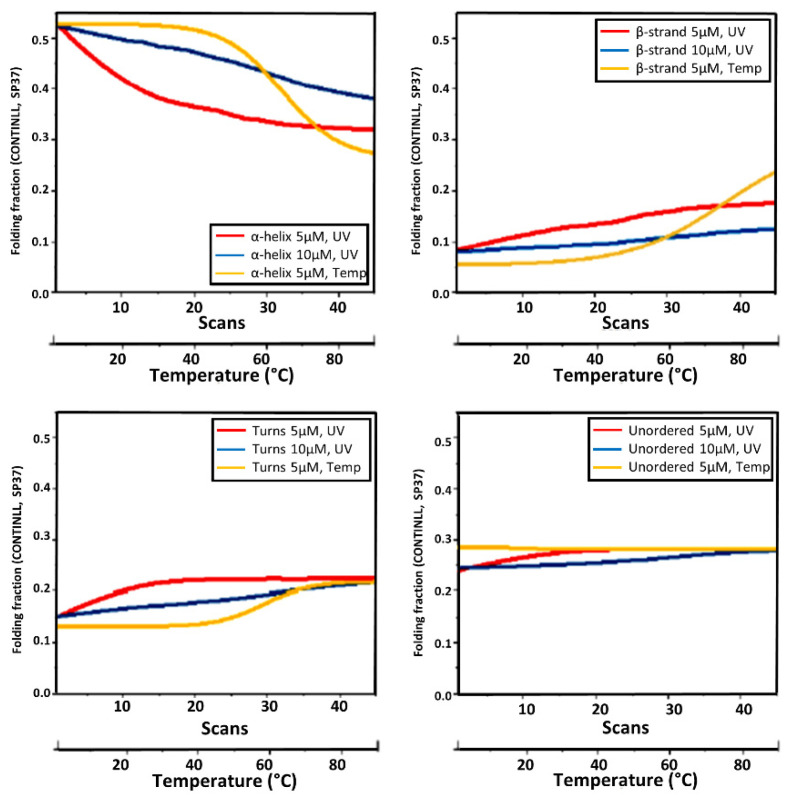
Fractions of elements of secondary structure: α-Helix (A), β-Strand (B), Turns (T) and Unordered (U) of HSAff as a function of UV (red for 5 µM and blue for 10 µM) and thermal denaturation (yellow for both concentrations) estimated from CD/SRCD spectra of Figure 3 using CONTINLL algorithm [17] with the SP37 data set from 37 soluble proteins.

**Figure 5 ijms-22-06512-f005:**
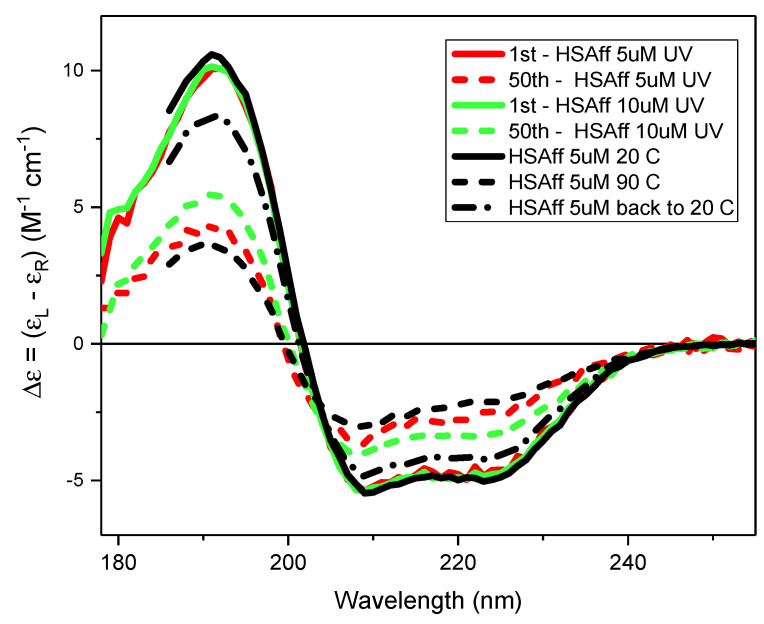
CD spectra in Δε unit of 5 µM identical to those of 10 µM of HSAff at 20 °C (solid blue), 90 °C (dashed blue), and back at 20 °C (dashed-dotted blue). First (solid) and 50th (dashed) SRCD spectra of HSAff 5 µM (pink) and 10 µM (green). The SRCD spectra measured after 24 h were identical to the 50th consecutive repeated scans of both 5 µM and 10 µM HSAff (spectra not shown).

**Figure 6 ijms-22-06512-f006:**
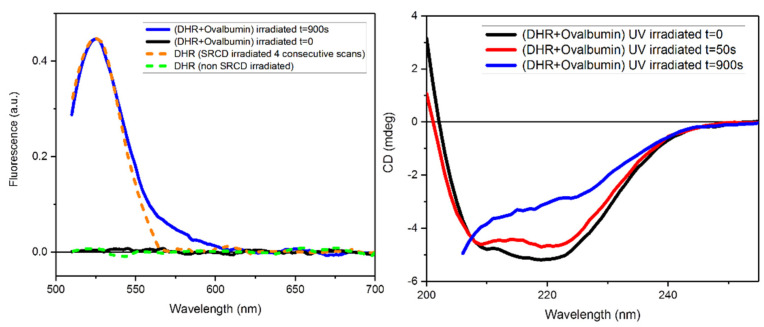
Characterization of the interaction between ovalbumin and DHR-123. (**Left**) Fluorescence emission spectra of DHR-123 (0.288 µM) in the presence of ovalbumin (0.009 mg/mL) irradiated with UV-C lamp in 20 mM PBS buffer, pH 7.4. Fluorescence spectra were recorded at 25 °C using a Chirascan Plus spectrometer, excitation at 505 nm, integration time 1 s, 1.0 cm cell (3000 µL), monochromator slit widths 4 nm. (**Right**) With the same set up but in CD mode the far-UV CD spectra of ovalbumin (0.009 mg/mL) in 20 mM PBS buffer, pH 7.4, were measured in the presence of DHR-123 (0.288 µM) both irradiated with a UV-lamp (t = 0 (black), t = 50s (red), and t = 900s (blue)). CD spectra were recorded at 25 °C using a Chirascan Plus spectrometer, integration time 1 s, 1.0 cm cell (3000 µL), monochromator slit widths 1.0 nm.

**Table 1 ijms-22-06512-t001:** Secondary Structure Estimation (SSE) from spectra of Figure 4, SRCD data of 5 µM and 10 µM HSAff for UV irradiation (1st and 50th spectra at 20 °C) and from CD data measured at 20 °C, 90 °C and cooled back to 20 °C. SSE was calculated using CONTINLL [17] with SP37 data set of 37 soluble proteins.

	% α-Helix (n° Helices *; aa Length)	% β-Strand (n° Strands *; aa Length)	% Turns
HSAff 5 µM and 10 µM, 1st at 20 °C	**53.0** (4.7; 11.2)	**8.4** (1.8; 4.7)	**14.8**
HSAff 5 µM, 50th at 20 °C	**31.4** (3.7; 8.5)	**18.1** (3.6; 5.1)	**22.0**
HSAff 10 µM, 50th at 20 °C	**37.4** (4.2; 8.9)	**12.3** (2.9; 4.2)	**22.2**
HSAff 5 µM and 10 µM at 90 °C	**26.6** (3.3; 7.9)	**23.7** (4.4; 5.4)	**21.5**
HSAff 5 µM and 10 µM, back to 20 °C	**45.8** (4.4; 10.5)	**8.8** (2.1; 4.3)	**15.7**

* The number of helices or strands is per 100 amino acids (aa).

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
