# Peer review of "Free Radicals and ROS Induce Protein Denaturation by UV Photostability Assay"

_ijms, 2021, doi:10.3390/ijms22126512_

Round 1

Reviewer 1 Report

Review of the manuscript ijms-1248477

Title: Free radicals and ROS induce protein denaturation by UV 2 photostability assay

Authors: Paolo Ruzza , Claudia Honisch, Rohanah Hussain and Giuliano Siligardi

Photostability of proteins is in the focus of the presented study. The thermal and UV denaturation studied using CD spectroscopy of the proteins was compared. It was shown that similarly to thermal denaturation UV denaturation varies from protein to protein, but interestingly is affected by ligand binding interactions. Consequently the mechanism of UV denaturation was proposed involving the local heating from irradiated protein bound water molecules or from the formation of free radical products. As the comparison of thermal and UV denaturation of human serum albumin showed different pathways the fluorescent probe allowing direct detection of the presence of the free radicals was employed. Thus combine used of CD and fluorescence spectroscopy elegantly showed that UV denaturation is not caused by local heating of the molecules of water bound to the protein but solely due to free radicals.

The paper is clearly written and straightforwardly argued. I would recommend the paper to be published, only I would prefer if author could specify in experimental part the source of used proteins.

Reviewer 2 Report

This paper reports synchrotron radiation cd spectroscopy of proteins that shows how proteins unfold due to the exposure to high-flux UV radiation. The basic result is that proteins unfold due to the generation of radicals in the aqueous media rather than simple thermal unfolding. This is not surprising but the data is clear and publishable anyways. 

My biggest problem with this paper is the disorganized way the data and discussion are presented. The introduction basically summarizes two very complex figures, and the discussion section (section 3) is non-existent (the authors didn't even bother to delete the text from the MDPI template). Within the results section (section 2), Figure 3 seems to squeeze too much information together. I think the authors should try to improve the organization of this manuscript.

Also I think that the data and experiments presented are not "apples-to-apples" comparisons. The use of DHR-123 should have been done with HSA, not ovalbumin. And the analysis is suspect also: The excellent analysis in Table 1 and Figure 4 should have been done with ovalbumin with DHR-123. The authors need to try to reorganize their discussion and results to explain why the HSA results are distinct from the ovalbumin results, or else do more analysis of the ovalbumin results and include a new figure (essentially Fig 4 but for ovalbumin) and table (Table 1 but for ovalbumin). I guess the essential question is, why did authors not do the same technique and analysis for both proteins? Why use one technique and analysis for HSA and then something quite different for ovalbumin?

Also the use of rate constants to quantify the unfolding process is not accurate, in my opinion. The way the experiments are done (essentially just scan over and over) does not translate to a true kinetics experiment, because the experiment that detects the structure is what is causing the protein unfolding, as I understand the authors' description of this method. A true kinetics experiment needs to separate the cause of the unfolding from the method that detects the unfolding. Qualitatively the authors can infer differences in the kinetics between different experiments (eg, unfolding "rates" in 5 uM vs 10 uM, etc) but any quantitative analysis of rate constants, that would be comparable to unfolding rate constants, is unacceptable. Perhaps the authors can use some new vocabulary to quantify the unfolding times but the use of rate constants implies some physically rigorous kinetics analysis that are not available from the experiments as described in this work.

In line with the above comment, actually authors COULD have reported rate constants for the ovalbumin work and those would have been at least some rigorous kinetics parameters, but they did not do any spectra vs time measurements. Why not?

The data presented in this work is unambiguous and certainly publishable but the discussion of the results is lacking and disorganized. The authors need to address my comments and revise the manuscript.
